# Regulation of Mitochondria-Derived Immune Activation by ‘Antiviral’ TRIM Proteins

**DOI:** 10.3390/v16071161

**Published:** 2024-07-19

**Authors:** Seeun Oh, Michael A. Mandell

**Affiliations:** 1Department of Molecular Genetics and Microbiology, University of New Mexico Health Sciences Center, Albuquerque, NM 87131, USA; seoh@salud.unm.edu; 2Autophagy, Inflammation and Metabolism Center of Biomedical Research Excellence, University of New Mexico Health Sciences Center, Albuquerque, NM 87131, USA

**Keywords:** autophagy, mitophagy, mitochondria, RIG-I, MDA5, cGAS, STING, TBK1, MAVS, tripartite motif, TRIM, antiviral defense, interferon, inflammation, restriction factor

## Abstract

Mitochondria are key orchestrators of antiviral responses that serve as platforms for the assembly and activation of innate immune-signaling complexes. In response to viral infection, mitochondria can be triggered to release immune-stimulatory molecules that can boost interferon production. These same molecules can be released by damaged mitochondria to induce pathogenic, antiviral-like immune responses in the absence of infection. This review explores how members of the tripartite motif-containing (TRIM) protein family, which are recognized for their roles in antiviral defense, regulate mitochondria-based innate immune activation. In antiviral defense, TRIMs are essential components of immune signal transduction pathways and function as directly acting viral restriction factors. TRIMs carry out conceptually similar activities when controlling immune activation related to mitochondria. First, they modulate immune-signaling pathways that can be activated by mitochondrial molecules. Second, they co-ordinate the direct removal of mitochondria and associated immune-activating factors through mitophagy. These insights broaden the scope of TRIM actions in innate immunity and may implicate TRIMs in diseases associated with mitochondria-derived inflammation.

## 1. Introduction

Over the last two decades, the tripartite motif-containing (TRIM) protein family has increasingly been linked to regulating innate immune responses to viruses [1] and other microbial pathogens [2,3,4,5]. TRIMs are characterized by their conserved domain structure that consists of an N-terminal RING domain that is involved in the conjugation of ubiquitin or ubiquitin-like proteins to substrates, one or two BBox domains, and an extended coiled-coil (CC) domain that mediates TRIM dimerization. Most TRIMs also have domains at their C-terminus that are thought to mediate interactions with their binding partners. Higher-order assembly or oligomerization of TRIMs is often essential for them to carry out their biological functions. The number and diversity of TRIMs has greatly expanded in vertebrates relative to other metazoans: whereas worms and flies have fewer than 20 TRIM genes, zebrafish have >200 TRIM genes [6]. The human genome encodes ~80 TRIM genes, often with multiple isoforms. With the high number of TRIM genes comes a broad diversity of functional roles for TRIM proteins. TRIMs’ enzymatic activity as E3 ligases is sometimes, but not always, essential for them to fulfill these roles. While various TRIMs function in the development or maintenance of homeostasis, actions in responding to and protecting against infection are a common feature of many TRIMs in both mammals and fish [6,7]. As a family, the expression of many TRIM proteins is increased in response to interferon α/β treatment or viral infection [8,9]. Functionally, some TRIMs can directly interfere with the life cycle of pathogens, as exemplified by the protein TRIM5α which can robustly protect cells from infection by certain retroviruses [10]. Alternatively, many TRIMs indirectly impact the outcome of infection by regulating innate immune-signaling pathways [9].

While TRIMs are under intensive study for their actions in immune defense against pathogens, less is known about their contributions to regulating sterile inflammatory processes. Mitochondria have emerged as important sources of sterile inflammation because they contain pathogen-like molecules that can serve as damage-associated molecular patterns (DAMPs) and trigger immune activation if released into the cytosol following mitochondrial damage. Additionally, mitochondria serve as membrane-bound platforms for innate immune signal transduction. Inappropriate mitochondria-related inflammation is increasingly linked to important neurodegenerative or autoimmune diseases [11,12]. TRIM proteins can control mitochondria-based immune responses in two ways. First, extensive studies have identified mechanisms through which TRIMs directly modulate the activity of immune signal transduction pathways that respond to molecules of viral and mitochondrial origin. Second, TRIMs with known roles in antiviral immunity have recently emerged as regulators of mitophagy, a pathway that can attenuate immune signaling by eliminating mitochondria and their associated immuno-stimulatory molecules.

## 2. Stimulation of “Antiviral” Responses by Mitochondria

Mitochondria are organelles that play essential roles in eukaryotic cells. In addition to their best known function of generating ATP through oxidative phosphorylation, mitochondria are key components of nearly all metabolic processes [13]. Mitochondria consist of two membranous structures: an outer membrane (OMM) that is in communication with the cytosol and other membranous structures, and a highly convoluted inner membrane (IMM) that houses the electron transport chain machinery. Between these two membranes is the inner-membrane space. The IMM encloses the mitochondrial matrix, which consists of metabolic enzymes and the mitochondrial genome. While most mitochondrial proteins are encoded by the nuclear genome, the mitochondrial genome encodes the genes for the electron transport chain machinery and for mitochondrial tRNAs and rRNAs.

Mitochondria likely evolved as part of an ancient mutualism between α-proteobacteria-related prokaryotes and proto-eukaryotes [13]. Mitochondria retain molecules and structures that reflect their evolutionary past and that are present in modern intracellular pathogens. When mitochondria are damaged, these molecules can be exposed, leading to activation of innate immune signaling. For example, cardiolipin is a phospholipid found in bacterial membranes and in the IMM (but not in other eukaryotic cell membranes). When exposed to the cytosol, cardiolipin binds directly to the inflammasome component NLRP3 and is sufficient to drive caspase-1 activation [14].

Mitochondrial nucleic acids can be potent ligands for several pattern-recognition receptors (PRRs) that induce classically “antiviral” interferon α/β responses. Upon mitochondrial damage, mitochondrial DNA (mtDNA) can be released into the cytosol through pores in the OMM generated by oligomerized Bax and Bak [15,16,17], or through oligomerization of the mitochondrial protein VDAC1 [18]. Accumulation of mtDNA in the cytosol can activate interferon expression by triggering the cGAS/STING pathway [19,20]. Mitochondrial DNA is a potent activator of the cGAS/STING pathway, at least partly due to the geometry imposed on mtDNA by binding to the mitochondrial transcription factor TFAM [21]. Increasingly, studies in model organisms have linked “antiviral-type” cGAS signaling induced by mtDNA to aging [22], auto-immunity [20], and neurodegenerative diseases [23]. mtDNA can also activate the endosomal DNA receptor TLR9 if it is released into the extracellular space [24]. The ability of mtDNA to trigger these PRRs or to activate the AIM2 inflammasome [25] is substantially increased in the absence of mitophagy, a degradative process that can clear the cytoplasm of damaged mitochondria and their associated molecules (see below).

Mitochondrial DAMPs (mtDAMPs) can also activate MDA5 [26] and PKR [27], both of which recognize double-stranded RNA (dsRNA). Although dsRNA is typically considered to be present only in cells infected with RNA viruses, this is not the case. Instead, bi-directional transcription of the circular mtDNA genome generates complementary RNA molecules that can dimerize by base pairing. However, the abundance of mitochondrial dsRNA is typically kept very low due to the actions of the enzymes SUV3 and PNPase, which mediate the degradation of the light (L) strand of mtRNA [26]. In the absence of PNPase, the cellular abundance of dsRNA dramatically increases, with some of it escaping into the cytosol via Bax-Bak pores, leading to type I interferon expression in a manner requiring MDA5 [26]. Inhibition of mitophagy was also reported to increase the abundance of dsRNA in cells and to activate PKR signaling [27].

Collectively, the studies highlighted above illustrate significant parallels between the antiviral immune response and the immune response elicited by mitochondria-derived molecules. As DAMPs, mtDNA and mtRNA can activate nucleic acid receptors, in a similar way to viral PAMPs, leading to the induction of interferon (IFN) through the same signal transduction pathways.

## 3. Mitochondria Participate in Antiviral Responses

In addition to stimulating antiviral-like responses through the release of DAMPs, mitochondria are also active players in cellular responses to viral infection. Mitochondria can contribute to antiviral signaling through the inducible release of mtDAMPs in response to viral infection or cytokine receptor stimulation. Although the cGAS/STING pathway triggers type I interferon production upon sensing cytosolic DNA, it is targeted by diverse RNA viruses that specifically down-regulate or otherwise disrupt this pathway [28,29,30]. This counter-intuitive activity suggests that DNA-sensing by cGAS/STING is of relevance to RNA viruses. Indeed, mtDNA is often released into the cytosol following viral infection to initiate or enhance the expression of antiviral genes and interferon production in a cGAS-dependent manner. In this way, cGAS can initiate antiviral responses to RNA viruses. In some cases, infection-induced mtDNA release contributes to protective antiviral responses [31,32,33]. However, mtDNA-induced activation of cGAS signaling is reported to contribute to COVID-19 disease following infection with SARS CoV-2 [34]. Additionally, treatment of cells with the pro-inflammatory cytokine IL-1β has been shown to trigger mtDNA release and consequent cGAS-STING-IRF3 signaling [35]. IL-1β treatment restricted Dengue virus infection in vitro in a manner requiring STING, thus demonstrating the functional importance of the IL-1β-mtDNA-cGAS-IRF3 axis [35]. Collectively, these findings establish mtDNA as a DAMP that is released in a programmed manner as part of innate immune responses.

Additionally, the mitochondrial antiviral signaling protein (MAVS, also known as IPS-1, VISA, or Cardif) is an essential signal transduction node downstream of the dsRNA sensors RIG-I and MDA5 [36,37,38]. RIG-I, MDA5, and the related protein LGP2 are collectively referred to as RIG-I-like receptors or RLRs. RIG-I and MDA5 both recognize dsRNA, albeit with different specificities [39]. Under normal conditions, these receptors are localized in the cytosol in an inactive state. However, dsRNA ligand binding induces a conformational shift in the receptor structure, exposing the CARD domains at the protein’s N terminus and allowing for oligomerization of the receptor. RIG-I or MDA5 oligomers are then translocated to the mitochondria where they can interact with, and activate, MAVS. MAVS is a transmembrane protein anchored in the outer mitochondrial membrane by its C terminus. At its N-terminus, MAVS has its own CARD domain. Through CARD–CARD interactions, RLR–MAVS complexes form a prion-like aggregate that promotes the recruitment, assembly, and activation of downstream signaling components including TNF receptor-associated factor (TRAF) family ubiquitin ligases. The actions of these ubiquitin ligases are required for the subsequent activation of kinases that phosphorylate and activate IRF-3 and NF-κB transcription factors, leading to the establishment of an antiviral state and the release of inflammatory cytokines. The assembly of the MAVS signalosome is tightly regulated, including by TRIM protein-mediated ubiquitination (as discussed below). Additionally, a number of mitochondrial resident proteins have been shown to interact with and either positively or negatively influence MAVS signaling [40].

Mitochondria can also serve as platforms for activation of the pro-inflammatory transcription factor NF-κB. While this can happen in response to mitochondrial damage [41], a recent study has shown that mitochondria are also key sites of NF-κB in response to cytokine signaling [42]. Activation of TNF receptors at the plasma membrane rapidly triggers the deposition of linear ubiquitin chains on mitochondria. These ubiquitin chains serve as activation platforms for the NEMO/IKK complex in a manner facilitated by the mitochondrial kinase PINK1 [42]. Activated IKK de-represses NF-κB [43]. In agreement with the concept of mitochondria being NF-κB activation sites, Wu et al. found that TNF treatment increased the recruitment of both NEMO and the NF-κB component p65 to mitochondria [42]. Interestingly, in this study the authors observed that TNF stimulation caused mitochondria to assume a perinuclear localization, thus increasing mitochondria–nucleus contact sites. They proposed that this relocalization of mitochondria allowed the organelles to serve as shuttles for active NF-κB to facilitate its nuclear import [42]. Whether mitochondria can also ferry other transcription factors like IRF-3 that are activated on mitochondrial platforms to the nucleus is unknown.

As sources of DAMPs, hubs for immune signaling, and masters of immuno-metabolism, mitochondria are central players in the antiviral response and in homeostasis. Regulating these activities is crucial to maintaining cellular and organismal health, as there is a need to maintain robust responsiveness to viral threats or potentially pathogenic mitochondrial damage, while at the same time avoiding over-exuberant or otherwise inappropriate immune activation. As described below, TRIM proteins have emerged as key regulators of mitochondria-based antiviral or “antiviral-like” immune responses.

## 4. Regulation of Innate Immune Signaling by TRIMs

*TRIM-mediated regulation of signaling pathways that can be activated by mtDAMPs.* In 2007, Gack et al. reported that TRIM25 played an essential role in RIG-I-dependent interferon production in response to viral RNAs [44]. The authors found that TRIM25 mediated the deposition of non-degradative K63-linked poly-ubiquitin chains on RIG-I. These ubiquitin chains were essential for RIG-I oligomerization and interactions between RIG-I and MAVS. This report was the first of many that have collectively established TRIM proteins as regulators of innate immune responses to viral PAMPs, particularly through the RLR and cGAS/STING pathways. As shown in Figure 1, TRIMs can impact each stage of these innate immune signal transduction cascades.

Since several published reviews discuss the many ways in which TRIMs regulate immune signaling in detail [6,45,46,47], here we will only provide a high-level overview of these mechanisms. As described above for TRIM25, this can include impacting the ability of multi-protein complexes to form. While this activity often involves the TRIM’s deposition of K63-linked poly-ubiquitination that can act as a scaffold for protein–protein interactions [44,48,49], in some cases the TRIM itself acts as a bridge between proteins in the complexes [50]. TRIM action can also involve altering the proteasomal or lysosomal degradation of signaling proteins or complexes, leading to changes in their overall expression level [51,52,53,54,55]. Additionally, several TRIMs have been shown to interact with transcription factors to alter the transcription of innate immune genes [56,57,58]. Finally, some TRIMs can attenuate or prevent innate immune activation by effectively eliminating the innate immune trigger. For example, the ability of retroviral infection to stimulate cGAS signaling in dendritic cells was reported to be inversely correlated with the ability of the cells to carry out TRIM5α-mediated retroviral restriction because the actions of TRIM5α prevented the accumulation of cGAS-detectable reverse transcription products [59]. It is important to note that all of the studies described above focused on TRIM actions in response to viral infection or model PRR ligands. While it is likely that the TRIMs play the same role when these pathways are stimulated by mtDAMPs, this concept has not been tested.

*Regulation of MAVS activation by mitochondria-localized TRIMs.* RLR signaling converges on the mitochondrial protein MAVS. Assembly and activation of the MAVS ‘signalosome’ requires MAVS aggregation and interactions with upstream RLR proteins and with downstream signaling factors such as IKK complexes and TBK1 [60]. These processes are highly regulated at the post-translational level, particularly involving modification by ubiquitin or ubiquitin-like molecules. MAVS has 14 lysine residues that can potentially be targeted by E3 ligases [61]. The impact of ubiquitin modification on MAVS activity depends on two factors: (1) which lysine residue in MAVS is ubiquitinated; and (2) which of the eight different ubiquitin chain linkages are used. TRIMs are capable of catalyzing the ligation of K11-, K27-, K48-, and K63-linked poly-ubiquitin chains on to other proteins. Some TRIMs have also been shown to act as E3 ligases that mediate the conjugation of SUMO, ISG15, or other ubiquitin-like proteins to substrate proteins [62,63]. As part of the ‘ubiquitin code’, these different modifications have differing impacts on protein function [64]. For instance, ubiquitination with K11- and K48-linked poly-ubiquitin chains predominantly results in proteasomal degradation [65], while K63-linked poly-ubiquitination can either stabilize protein–protein interactions or be a target of autophagy-based degradation [66].

Several TRIM proteins localize to mitochondria where they function as ubiquitin ligases promoting MAVS activity by enabling the assembly of MAVS signalosomes. Other mitochondrial-localized TRIMs can act to attenuate MAVS signaling by promoting its proteasomal degradation. Figure 2 summarizes the ubiquitin modifications of MAVS that are mediated by TRIMs and how these modifications impact MAVS signaling. At least 11 TRIMs interact with MAVS based on published studies and NCBI databases; these include TRIMs 7, 14, 19, 21, 25, 28, 29, 31, 40, 44, and 67.

The inducible formation of prion-like MAVS aggregates is required to transduce RLR signaling [60]. The formation of MAVS aggregates requires the polymerization of MAVS and RLR CARD domains [36]. Thus, stabilization of MAVS–RLR interactions will enhance MAVS aggregation and downstream signaling. Liu et al. demonstrated that MAVS aggregation requires the actions of TRIM31 [61]. While TRIM31 primarily localized to the cytosol in uninfected cells, Sendai virus infection relocalized TRIM31 to mitochondria, where TRIM31 interacted with MAVS and mediated its K63-linked polyubiquitination at K10, K311, and K461. Replacing these lysine residues with arginine reduced TRIM31-mediated MAVS ubiquitination and impaired both MAVS aggregation and virus-induced MAVS signaling. In addition to directly ubiquitinating MAVS, TRIM31 was also reported to generate unanchored K63-linked polyubiquitin chains that associated with MAVS [67]. Either attached or unattached ubiquitin chains that are associated with MAVS serve as secondary scaffolds securing MAVS interactions with ligand-bound RLR complexes [61,67].

TRIM14 also promotes the assembly of active MAVS–RLR complexes. Unlike most TRIM proteins, TRIM14 lacks a RING ubiquitin ligase domain. Upon viral infection, TRIM14’s mitochondrial localization and MAVS interactions are increased [50,68]. Subsequently, TRIM14 forms a complex with two additional proteins: WHIP and PPP6C [50]. WHIP contains an ubiquitin-binding zinc-finger domain that binds to K63-ubiquitinated RIG-I. Thus, the TRIM14-WHIP complex serves as an adaptor bridging MAVS and RIG-I. PPP6C is a phosphatase that removes two inhibitory phosphate modifications from RIG-I, potentiating RIG-I’s signaling capacity [50].

TRIMs are also reported to facilitate the interaction between MAVS and its downstream signaling factors. In addition to the actions of TRIM14 mentioned above, TRIM14 also serves as an adaptor between MAVS and the protein NEMO (NF-κB essential modifier), a crucial component of the IKK complex that promotes activation of TBK1 and NF-κB [68]. TRIM21 may have a similar effect. RNA virus infection increases interactions between interferon-inducible TRIM21 and MAVS. TRIM21 then promotes K27 poly-ubiquitination of MAVS at K325, a modification which stabilizes interactions between MAVS and TBK1 [69].

The abundance of MAVS protein is also an important regulator of MAVS function. In contrast to the TRIMs detailed above, several TRIMs with mitochondrial localization catalyze degradative ubiquitination of MAVS [70,71,72,73]. TRIM25 and TRIM28 carry out the K48-linked ubiquitination of MAVS, albeit at different sites, with TRIM25 ubiquitinating K7 and K10 [73], and TRIM28 ubiquitinating MAVS at K7, K10, K371, K420, and K500 [71]. TRIM29 induces degradation of MAVS via K11-linked poly-ubiquitination at the K371, K420, and K500 sites [72]. In the case of TRIM7, TRIM28, and TRIM29, MAVS degradation was shown to inhibit RLR signaling [70,71,72]. However, it should be noted that a TRIM’s ability to mediate MAVS degradation does not imply that the TRIM lacks antiviral activity. For instance, TRIM7 directly restricts certain noroviruses, flaviviruses, enteroviruses, and coronaviruses [74,75,76,77]. Additionally, TRIM25’s ability to promote proteasomal degradation of MAVS was positively associated with antiviral signaling [73].

Unlike the TRIMs detailed above, TRIM44 can enhance RLR signaling by preventing or reversing the K48 ubiquitination of MAVS, thus protecting MAVS from proteasomal degradation [78]. TRIM44 is reported to have deubiquitinase activity [79,80], and so it is possible that TRIM44 promotes RLR signaling by catalyzing the removal of degradative ubiquitin modifications deposited by other TRIMs.

## 5. TRIM-Mediated Mitophagy Regulates Mitochondrial “Antiviral-like” Signaling

Mitophagy has emerged as a new mode whereby TRIM proteins regulate mitochondria-associated antiviral signaling. Mitophagy is a homeostatic autophagy-based mechanism in which entire mitochondria or mitochondrial fragments are sequestered within a double-membraned vesicle, the autophagosome [81]. Autophagosomes fuse with lysosomes, resulting in the degradation of the autophagosome’s inner membrane and its luminal contents (Figure 3, top). Mitophagy is essential for maintaining mitochondrial quality control by eliminating damaged, dysfunctional, or otherwise unwanted mitochondria [82]. It also effectively reduces the abundance of mitochondrial DAMPs (e.g., mtDNA) that would otherwise serve as triggers for immune signaling and can degrade mitochondrial-associated immune signaling molecules (e.g., MAVS or NLRP3 inflammasome), leading to their silencing [12,81,83]. Three recent studies have shown that TRIM5α and TRIM27 localize to mitochondria and play key roles in initiating mitophagy [84,85,86].

TRIM5α was the first TRIM associated with antiviral defense and has been under intensive study. In 2004, Stremlau et al. found that expression of rhesus TRIM5α could render human cells strongly resistant to infection by HIV-1 [87]. Recent studies have extended TRIM5α’s antiviral actions to certain flaviviruses [88] and poxviruses [89], collectively establishing TRIM5α as a broadly acting, multifunctional antiviral protein. Interestingly, GWAS studies have indicated significant associations between variants of the *TRIM5* gene and several conditions that are not obviously connected with viral disease, including coronary artery disease [90,91,92,93], altered cholesterol levels [94,95,96,97,98], and multiple sclerosis [99], implying that TRIM5α may also play homeostatic functions.

While TRIM5α had already been linked to autophagy [100,101,102], the first indication that it played roles in mitophagy came from a study of TRIM5α’s interacting partners. Proteomic analysis revealed that TRIM5α is in close proximity to a large number of mitochondrial proteins [85]. Among these, several have roles in marking damaged mitochondria for mitophagy-based elimination. These mitophagy ‘eat-me’ tags included NIPSNAP1/2, Prohibitin 2, SAMM50, and FKBP8. Mitochondrial damaging agents that induce mitophagy increased interactions between TRIM5α and ‘eat-me’ tag proteins. Importantly, cells lacking intact copies of the *TRIM5* gene were defective in mitophagy induced by a set of mitochondrial poisons [85,86]. The mitophagy deficit in *TRIM5* knockout cells was accompanied by an increase in the expression of antiviral (IFIT1, OAS, and phosphorylated IRF-3) and inflammatory proteins (pro-IL-1β) following treatment with the mitochondrial-damaging anthelminthic drug ivermectin [85]. This result fits a model in which TRIM5α-dependent mitophagy restrains immune activation following mitochondrial damage, but further experiments are required to confirm that mitophagy defects underlie this phenotype.

Mitophagy involves the sequential and often self-amplifying recruitment and activation of dozens of proteins that serve to: (1) flag damaged mitochondria for recognition by the autophagy machinery; (2) recruit autophagy initiation factors; (3) activate the generation of phosphatidyl inositol 3-phosphate enriched membranes at the autophagosome initiation site; (4) fabricate the autophagosome membrane around the damaged mitochondria; and (5) fuse the autophagosome with lysosomes [103]. TRIM5α was found to act at a very early step stage in this process, and was responsible for the recruitment of the most of the upstream members of the core autophagy machinery to damaged mitochondria in a manner requiring its ubiquitin ligase activity [85]. Much of TRIM5α’s mitophagy action is attributable to its effects on the kinase TBK1 [86]. Like TRIM5α, TBK1 is best known for its functions in innate immunity, where it phosphorylates IRF-family transcription factors downstream of TLR, RLR, or cGAS/STING signaling [104]. However, TBK1 also plays key roles in mitophagy initiation [105,106,107]. TRIM5α catalyzes the K63-linked ubiquitination of TBK1 at multiple lysine residues, initiating a pro-mitophagy feed-forward cycle that assembles TBK1 and autophagy adaptor proteins on damaged mitochondria (Figure 3, middle) [86]. Activation of this cycle promotes the downstream assembly of the rest of the mitophagy machinery, as TBK1 can phosphorylate autophagy proteins to increase their affinity for autophagy substrates [108] or to promote their assembly and activation [105,109,110,111,112]. By activating TBK1, TRIM5α is upstream of all of these processes (Figure 3, bottom). Given TRIM5α’s well-known functions as a PRR for viral structures and its newly uncovered ability to respond to mitochondrial damage by activating TBK1, TRIM5α could be considered as a PRR for damaged mitochondria, with mitophagy being the output. Identification of the mitochondrial ligand(s) that are recognized by TRIM5α to initiate mitophagy will be an important next step.

TRIM27 (also known as Ret finger protein or RFP) was investigated for a role in autophagy based on the observations that TRIM27 colocalized with the autophagy receptor protein p62 and that TRIM27 over-expression altered mitochondrial morphology and sub-cellular localization [84]. Further experiments showed that TRIM27 deficiency reduced mitophagy stimulated by the exogenous expression of the autophagy protein LC3A and the mitophagy ‘eat-me’ tag FKBP8 [84]. TRIM27-dependent mitophagy required TBK1, with TRIM27 expression enhancing the recruitment of active TBK1 to mitochondria [84]; a finding reminiscent of TRIM5α’s actions in mitophagy [86]. While the impact of TRIM27-depedent mitophagy on immune signaling was not tested, multiple studies have implicated TRIM27 in suppressing antiviral signaling in a variety of contexts [113,114,115].

While deficiency of either TRIM5α or TRIM27 is sufficient to disable mitophagy under certain conditions, both TRIMs appear to have similar TBK1-focused mechanisms. Indeed, over-expression of TRIM27 could rescue mitophagy in TRIM5-knockout cells [86], suggesting that the two TRIMs have some degree of mechanistic redundancy. However, other data suggest that TRIM5α and TRIM27 may co-operatively execute mitophagy, as TRIM27 expression increased TRIM5α’s interactions with TBK1 [86]. Further investigation is necessary to determine how these two TRIMs co-operate in mitophagy, under what physiological conditions TRIM5α- or TRIM27-dependent mitophagy are activated, and how TRIM-dependent mitophagy regulates the level of cellular immune activation.

Whether other TRIMs in addition to TRIM5α and TRIM27 can directly activate the mitophagy machinery is not known. However, TRIM28/KAP1 was shown to be essential for mitophagy in erythropoiesis, where TRIM28 promoted the expression of multiple mitophagy-related genes at a transcriptional level [116]. Additionally, ultrastructural analysis of cardiac tissue harvested from TRIM65- or TRIM76/Myospryn-knockout mice revealed the presence of swollen and vacuolated mitochondria [117,118], a phenotype that might be expected when mitophagy is dysfunctional. These findings hint that mitophagy regulation may emerge as a common feature of multiple TRIMs. A key unresolved question is how TRIM proteins’ mitophagy functions interrelate with their activities in antiviral defense. Several studies have shown that mitophagy favors virus production because it can attenuate interferon responses to infection [119,120,121,122,123,124,125], and so TRIM-mediated mitophagy may counteract TRIM-mediated antiviral defenses. Addressing this question will require further investigation.

## 6. Conclusions

In this review, we have discussed how TRIMs, which are generally associated with antiviral immunity, can regulate mitochondria-based immune reactions. TRIMs are positioned to accomplish this indirectly by modulating signaling that is responsive to mtDAMPs or by directly acting on mitochondria via modification of MAVS activity or through mitophagy-based elimination of mitochondrial molecules. Given that TRIMs are increasingly found to be associated with mitochondria, it is possible that they may exert additional effects on the organelle, resulting in changes in mitochondrial morphology and dynamics, metabolism, or apoptotic signaling (Table 1). Mitochondrial defects are linked to a wide variety of human diseases and to the aging process [13], and thus it seems likely that future research will implicate TRIMs as important factors in diseases of mitochondrial origin.

## Figures and Tables

**Figure 1 viruses-16-01161-f001:**
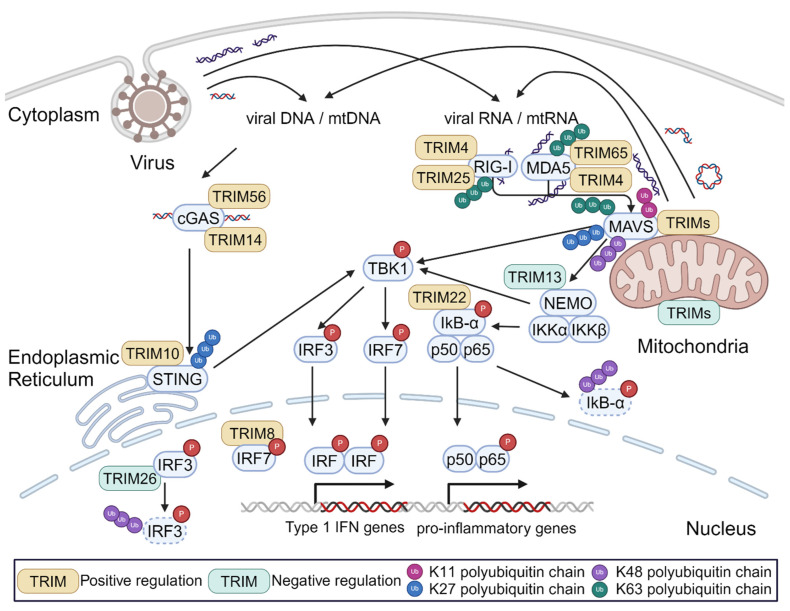
Regulation of TRIMs in innate immune signaling stimulated by viral PAMPs or mitochondrial DAMPs. When exposed to the cytosol, both viral PAMPs (e.g., viral DNA, dsRNA, etc.) and mitochondrial DAMPs can activate the same immune signal transduction pathways that are extensively regulated by TRIM proteins.

**Figure 2 viruses-16-01161-f002:**
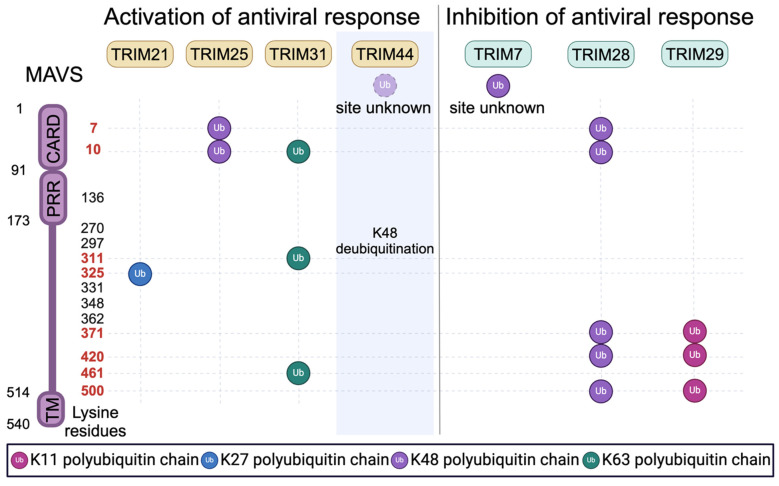
Regulation of MAVS signaling and stability by TRIM-mediated ubiquitination. MAVS contains 14 lysine residues distributed across its different domains, which include a single CARD domain, a proline-rich region (PRR), and a transmembrane domain (TM). Numbers in red indicate the target residues modified by TRIM-mediated ubiquitination. TRIM44 is a deubiquitinase that can stabilize MAVS by removing K48-linked polyubiquitin chains. TRIM7 is reported to induce K48-linked polyubiquitin chains, but the sites of TRIM7-mediated MAVS ubiquitination were not determined.

**Figure 3 viruses-16-01161-f003:**
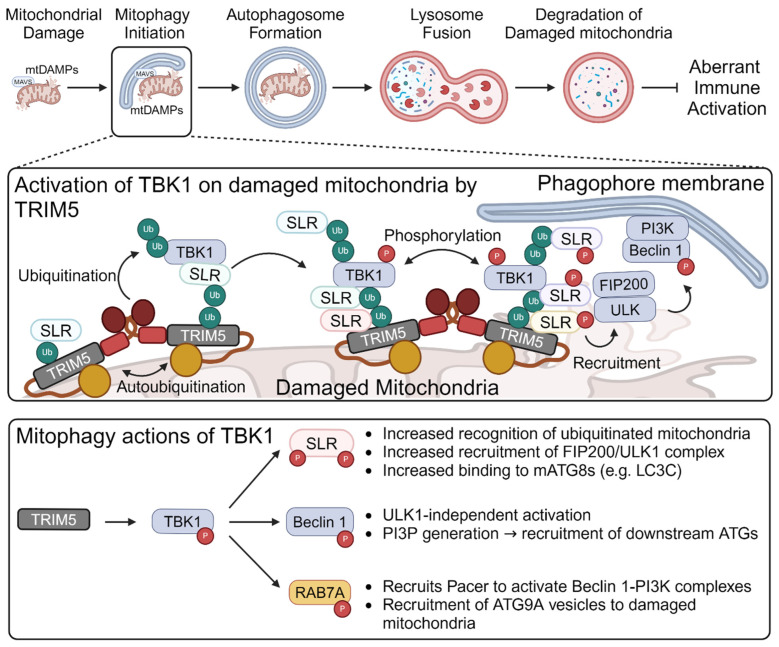
TRIM5-mediated mitophagy regulates immune activation. (**Top**), schematic of mitophagy pathway. Unwanted or damaged mitochondria and associated immunostimulatory molecules (e.g., mtDNA) or signaling factors (e.g., MAVS) are sequestered within an autophagosome and subsequently degraded following autophagosome/lysosome fusion. Elimination of mtDAMPs or signaling factors can prevent or attenuate mitochondria-based immune activation. (**Middle**), mechanism of TRIM5-mediated mitophagy. Mitochondrial damage activates TRIM5’s ubiquitin ligase activity. Active TRIM5 associates with TBK1 via shared interactions with Sequestosome-like receptors (SLRs, e.g., Optineurin, NDP52, TAX1BP1, p62, and NBR1). TRIM5 then ubiquitinates TBK1, allowing for the assembly of TRIM5-SLR-TBK1 complexes on damaged mitochondria in a feed-forward manner. TBK1 is concentrated in these complexes, enabling it to become activated by *trans*-autophosphorylation and carry out mitophagy functions. (**Bottom**), reported roles of TBK1 in mitophagy. TRIM5 promotes TBK1 activity in mitophagy. TBK1 can then phosphorylate SLRs, Beclin 1, and RAB7A. Phosphorylation of SLRs increases their ability to bind to ubiquitin chains on damaged mitochondria and increases the ability of SLRs to associate with other autophagy factors on mitochondria. Beclin 1 is a component of the class III phosphatidylinositol 3-kinase (PI3K) complex, and its phosphorylation at Ser15 enhances the complex’s activity to generate PI3P, a requirement for the recruitment of downstream autophagy factors to the autophagosome initiation site. RAB7A is essential for mitophagy. Its activation by TBK1 is important for the recruitment of the pro-mitophagy protein Pacer into Beclin 1 complexes, which takes the place of the autophagy-inhibitory Rubicon protein. RAB7A also recruits ATG9A vesicles to the autophagy initiation site in a manner requiring TBK1.

**Table 1 viruses-16-01161-t001:** Multiple TRIMs have reported mitochondrial localization and activities.

TRIMs	Synonym	Mitochondrial Functions
TRIM4		Regulates mitochondrial membrane potential and ROS production in response to oxidative stress [126].
TRIM5		Regulates mitophagy and inflammation [85,86].
TRIM7	RNF90/GNIP	Promotes MAVS degradation through K48 ubiquitination [127].
TRIM14		Enhancing RLR signaling via interacting with NEMO and RIG-I [50,68].
TRIM16	EBBP	Regulates mitochondrial membrane potential and apoptosis [128].
TRIM17	TERF	Regulates apoptosis via regulating the stability of apoptotic proteins [129,130,131].
TRIM19	PML	Regulates mitochondrial metabolism, induces apoptosis, inhibits inflammation [132,133,134].
TRIM21	SSA/RO52	Induces K27 ubiquitination of MAVS enhancing TBK1-IRF3 activation [69,135].
TRIM24	TIF1α	Regulates antiviral response via TRAF3 ubiquitination enhancing MAVS interaction [136].
TRIM25	EFP	Promotes MAVS degradation through K48 ubiquitination [44,73].
TRIM27	RFP	Induces mitophagy [84].
TRIM28	TIF1β/KAP1/KRIP-1	Promotes MAVS degradation through K48 ubiquitination, regulates apoptosis, mitophagy, and mitochondrial fusion [71,116,137].
TRIM29	ATDC	Promotes MAVS degradation through K11 ubiquitination [72].
TRIM31	RING	Induce MAVS aggregation and activation; proteasomal degradation of VDAC1. [61,67,138].
TRIM32	BBS11/HT2A	Regulates mitochondrial membrane potential and ROS production in response to oxidative stress. Regulates apoptosis and antiviral response [139,140,141].
TRIM34	IFP1	Regulates apoptosis [142].
TRIM39	TFP	Regulates apoptosis [143].
TRIM40	RNF35	Interacts with MAVS [53].
TRIM44		Stabilizes MAVS by deubiquitylation [78].
TRIM63	MURF1	Regulates mitochondrial metabolism [144,145,146].
TRIM65		May regulate mitophagy/apoptosis [117].
TRIM67	TNL	Interacts with MAVS [147].
TRIM72	MG53	Regulates mitophagy and inflammation [148].

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
