# Peer review of "Regulation of Mitochondria-Derived Immune Activation by ‘Antiviral’ TRIM Proteins"

_viruses, 2024, doi:10.3390/v16071161_

Round 1

Reviewer 1 Report

Comments and Suggestions for Authors

Viruses Review (7.1.24)

Regulation of mitochondria-derived immune activation by “antiviral” TRIM proteins

Post-translational modifications are an important means for cellular regulation of diverse biological functions including protein homeostasis, cellular signaling, and antiviral innate immunity. The TRIM family of E3 ubiquitin ligases have gained popularity in recent decades for being among the most prolific and characterized regulators of the immune response.

Oh and Mandell have assembled a review describing the recent developments of TRIM proteins involved in mitochondrial innate immunity. The resulting meta-analysis will be of broad interest to researchers studying the regulatory mechanisms of mitochondria in innate immune signaling.

The submission is well-written and would benefit from the addition of a few references that would clarify some sections. Additionally, a few typos were identified during the review for the authors to correct.

Specific minor revisions: Please incorporate the following revisions/references into their respective written sections.

1.     The authors make the following statement in section #2: Stimulation of “antiviral” responses by mitochondria:

“Unlike nuclear DNA, mitochondrial DNA is a potent activator of the cGAS/STING pathway…”.

However, nuclear DNA is a potent activator of cGAS/STING when it accumulates in the cytosol after DNA damage, see following references:

Li T, Chen ZJ. The cGAS-cGAMP-STING pathway connects DNA damage to inflammation, senescence, and cancer. J Exp Med. 2018;215(5):1287-1299. doi:10.1084/jem.20180139

Dunphy G, Flannery SM, Almine JF, et al. Non-canonical Activation of the DNA Sensing Adaptor STING by ATM and IFI16 Mediates NF-κB Signaling after Nuclear DNA Damage. Mol Cell. 2018;71(5):745-760.e5. doi:10.1016/j.molcel.2018.07.034

Please rephrase this sentence to acknowledge the immunostimulatory ability of host DNA and include the above references.

2.     There is a typo at the end of section #3: “This counter-intuitive activity suggests that DNA-sensing by cGAS/STING of relevance to RNA viruses.”

Correct to “cGAS/STING is of relevance…”

3.     There is a truncated sentence in section #4: “Figure 2 summarizes the ubiquitin modifications of MAVS that are mediated by TRIMs and how these modifications impact MAVS signal-

The end of this sentence appears cut off/missing before moving on to figure #2 and the next paragraph.

Author Response

We appreciate the reviewer's efforts and comments, which we feel have strengthened the manuscript. Please see the attached document for our responses to both reviewers' critiques. 

Reviewer 2 Report

Comments and Suggestions for Authors

The review by Seeun Oh and Michael A Mandell provides a comprehensive overview of TRIM proteins involved in regulation of mitochondria-related immune regulation.

Overall, this is an interesting and timely review, and refreshingly different from the myriads of TRIM reviews covering ‘TRIMs in the regulation of innate immunity’. It is also nicely highlighting mitochondria as – maybe still a bit underappreciated - central players in innate immune signaling, both their anti-viral functions as well as their role in sterile inflammation.

I only have a few minor comments for consideration:

TRIMs do not only dimerize via the CC domain but can also form higher order assemblies, would adjust the description.

During the evolution of TRIMs their number/diversity did not really expand from zebrafish to humans, I would suggest to rephrase that paragraph slightly.

In line “As a family, TRIM expression is increased in response to interferon α/β treatment or viral infection” – Not all TRIMs are ISGs, this should be corrected.

‘Despite the bacterial origin’ – Why ‘despite’, would their origin not suggest that they can be detected by PRRs ?

In line “Mitochondria have emerged as important sources of sterile inflammation because they contain pathogen-like molecules that can trigger immune activation if released into the cytosol.” – please refer here already to the pathogen-like molecules as DAMPs.

In line “Activated IKK de-“ – ‘the authors’ are referred to, it is not clear who this is, please rephrase. Of note, there are a few instances like this throughout the paper. Please consider to either name the authors or refer to ‘the authors’ if it is absolutely clear which citation is meant or whether it refers to the authors of this review.

I wonder whether mitochondria should be really dubbed as active players, if one of their main functions is to be a rather passive anchor for protein assemblies.

In line “The inducible formation of prion-like MAVS aggregates is required to transduce RLR signaling.” – please add a reference to this and the following sentence

Section starting with “Another ways in which mitochondria” could be moved further up.

The authors could consider to briefly (2 or 3 sentences) introduce the different types of Ub linkages and their function.

Figure 2 could be improved, as it is currently a bit confusing.

·        It would be great if the figure could be more table-like to help understand its purpose and have less ‘white space’.

·        Are all TRIMs mentioned in the figure also mentioned with their respective function/ub sites in the manuscript? I am missing the positive role of TRIM25, TRIM27 and TRIM44. The negative role of TRIM25 as mentioned in the text is not in the figure.

·        I wonder whether MAVS degradation is the only ‘negative’ role of TRIMs, and whether MAVS signalosome assembly should be only represented by PRRs/MAVS omitting the downstream factors that are mentioned in the text.

·        Activation should be before deactivation of the response, ie please switch the two sides of the figure.

·        Please highlight the 8 residues which are modified by TRIM-mediated ubiquitination in a different colour then the lysine residues in general.

The section “TRIM5a was the first…” should be streamlined, as it is not describing a TRIM function in or at mitochondria. Plus there is so much information on TRIM5a already out there. Mitophagy could be introduced earlier in the mitophagy paragraph.

In Conclusion: “TRIMs are positioned to accomplish this indirectly by modulating signaling that is responsive to mtDAMPS or by directly...“ – Please change DAMPS to DAMPs

Considering that there are over 80 TRIM proteins known, only three were discussed in more detail in this review. I wonder whether some TRIMs, which are currently not mentioned in the text and are related to immunity/mitochondria as listed in Table 1, should also be briefly described as well? Furthermore, some TRIMs that are currently only mentioned may deserve some more attention, e.g. TRIM7.

Author Response

(The authors gave the same response as above.)

Round 2

Reviewer 1 Report

Comments and Suggestions for Authors

No further comments.

Reviewer 2 Report

Comments and Suggestions for Authors

All my issues have been addressed. Thanks!